# Photoplethysmography behind the Ear Outperforms Electrocardiogram for Cardiovascular Monitoring in Dynamic Environments

**DOI:** 10.3390/s21134543

**Published:** 2021-07-02

**Authors:** Brian S. Bradke, Tiffany A. Miller, Bradford Everman

**Affiliations:** Spotlight Labs, 2 Kings Highway West Suite #104, Haddonfield, NJ 08033, USA; tiffany.miller@spotlightlabs.com (T.A.M.); everman@spotlightlabs.com (B.E.)

**Keywords:** SPYDR, physiological monitoring, photoplethysmography (PPG), electrocardiogram (ECG), motion artifact, pulse oximetry, hypoxia, blood oxygen saturation, pulse rate (PR), noninvasive monitoring

## Abstract

An increasing proportion of occupational mishaps in dynamic, high-risk operational environments have been attributed to human error, yet there are currently no devices to routinely provide accurate physiological data for insights into underlying contributing factors. This is most commonly due to limitations of commercial and clinical devices for collecting physiological data in environments of high motion. Herein, a novel Photoplethysmography (PPG) sensor device was tested, called SPYDR (Standalone Performance Yielding Deliberate Risk), reading from a behind-the-ear location, specifically designed for high-fidelity data collection in highly dynamic high-motion, high-pressure, low-oxygen, and high-G-force environments. For this study, SPYDR was installed as a functional ear-cup replacement in flight helmets worn by rated US Navy aircrew. Subjects were exposed to reduced atmospheric pressure using a hypobaric chamber to simulated altitudes of 25,000 feet and high G-forces in a human-rated centrifuge up to 9 G acceleration. Data were compared to control devices, finger and forehead PPG sensors, and a chest-mounted 12-lead ECG. SPYDR produced high-fidelity data compared to controls with little motion-artifact controls in the no-motion environment of the hypobaric chamber. However, in the high-motion, high-force environment of the centrifuge, SPYDR recorded consistent, accurate data, whereas PPG controls and ECG data were unusable due to a high-degree-motion artifacts. The data demonstrate that SPYDR provides an accurate and reliable system for continuous physiological monitoring in high-motion, high-risk environments, yielding a novel method for collecting low-artifact cardiovascular assessment data important for investigating currently inaccessible parameters of human physiology.

## 1. Introduction

Safe conduct during high-risk activities in extreme environments, such as those within tactical military flight, are highly dependent on the mental, physical, and emotional wellness of the personnel involved, yet there are currently no devices capable of routinely and reliably quantifying physiological and cognitive performance, nor warning the user of potential decremental dangers [1,2,3]. While tighter regulations, mandatory training, personal protective equipment, and “how do I feel” questionnaires are becoming commonplace, these methods fail to capture the individual nature of human factor risks involving the relationship between physiology and environment in hazardous environments [4,5,6]. As a result, they are not reducing the incidence or severity of human error-related accidents in high-risk professions [3,7,8].

Elucidating the specific contributing factors to operational mishaps, accidents, or deaths in extreme environments has been impeded the most by a lack of accurate and reliable methods for collecting physiological data in their often high-motion or high-force environments [8,9,10]. For example, in tactical flight, reduced atmospheric pressure in high altitudes, high and varied G-forces, and high motion are all extreme environmental conditions known to exert a large toll on normal operations of many physiological systems [11,12,13]. An additional limiting factor is that current sensors do not include a means of analyzing the data in real-time and alerting the user, or a third party, of crossing predetermined thresholds. Increasing our currently limited understanding of human physiology in highly dynamic environments with novel technologies capable of overcoming the challenges is a logical and important first step.

A means of effectively measuring human physiological factors, such as blood oxygen saturation (SpO2) and pulse rate (PR), plus analyzing the data and for decrement alerts, would allow us to more accurately assess and mitigate risk factors associated with human performance [14]. The advancement of sensing and wireless communication technologies has made it possible to develop systems to monitor these physiological activities continuously. Most of today’s commercial physiological monitoring wearable devices employ photoplethysmography (PPG) sensors which detect and analyze differential transmissivity of near-infrared light to calculate SpO2 and PR, Electrocardiogram (ECG) has been the gold standard of Heart Rate (HR) measurement, but it is primarily used and most effective in non-motion environments (e.g., lying still) [15]. ECG has been found to be highly unreliable and mostly unusable during periods of motion [16]. This is due to several factors and highlights the general unacceptability of using ECG in extreme operational environments. ECG technology also captures data through electrodes that require an additional input station, which are too cumbersome and distracting to wear while working, especially in a high-stakes environment such as flight, defense, or first response.

Wearable sensors are a common tool for detecting abnormal and/or unforeseen situations by monitoring human physiological parameters, such as SpO2 and PR [17]. However, the extreme operational environment of flight, especially in tactical military aircraft, where most dangerous or deadly physiological episodes occur, presents many challenges for the devices and techniques currently available for clinical use. SpO2 and PR are among the most common monitorable physiological parameters capable of determining a plurality of physical characteristics and ailments, including whether an individual is on the verge of losing consciousness or impaired cognitive functioning [11,18,19]. However, PPG sensor performance in the presence of motion varies widely among devices and manufacturers because of variations which occur in hardware, software, testing, and calibration methods, as well a number of problems from different measurement locations all over the body [20,21]. Fingers are one of the most common PPG sensor location because of the ease of use in clinical settings and high signal amplitude that can be achieved [20]. However, this configuration is not well suited to longer-term continuous, routine sensing because many activities involve use, or high movement, of the fingers, especially crucial for operators in extreme environments [10]. Typical PPG sensors are so sensitive to motion, that even the small movements of fingers have been shown to create signal artifacts which reduce the accuracy of the readings [14,22].

Additionally, in high-acceleration environments that are most extremely encountered for humans in piloting a fighter jet or simulated in a human centrifuge, peripheral sensors cannot be used effectively because of blood pooling and altered blood pressure at the extremities [22]. Blood pressure at peripheral sites is influenced by physiological events and body position variation, resulting in variances to the reading of the PPG measurement [22]. Vertical acceleration causes blood pooling to occur in the lower extremities, decreasing the venous return and altering the blood flow toward the brain. This creates inconsistency between the PPG’s measured values at the peripheral site and the physiological performance at the heart and brain, which are typically of greater concern [23].

As motion artifact is a well-known impediment for high-fidelity PPG sensor readings, signal noise reduction by using either signal processing algorithms or physical adaptations to sensor application, location, and overall system design is highly sought after [24]. The effect of several sensor design variables compounded with each other, such as sensor weight, relative motion, placement, and contact force against the skin, can further reduce motion artifact and increase accuracy and reliability of the system [14,25]. Additionally, measuring blood oxygenation directly to the brain, as opposed to extremities, is an important consideration in high-risk environments, where split-second decisions can be critical in preventing accidents.

Based on the need in many high-risk industries, especially tactical military flight, and the current state of the field for routine monitoring of operators, this study seeks to validate the accuracy of a human centrifuge, of a novel multiwavelength reflectance PPG sensor platform, called SPYDR^®^ (Standalone Performance Yielding Deliberate Risk), specifically designed to accurately measure and withstand the most rigorous physiological monitoring environments. SPYDR was previously shown to be as accurate as commonly approved finger and forehead PPG sensors through the gold-standard Arterial Blood Gas (ABG) testing used in accuracy approval of novel PPG sensors [26].

SPYDR was optimized to accurately recording human performance data in real time in high-acceleration, high-motion, high-altitude environments, while meeting challenges for fit, ease of use, and comfort. In addition, SPYDR’s on-board processing provides immediate detection and warning of potentially dangerous conditions, such as hypoxic incapacitation. An organic aural alerting system through bone-conduction technology provides an immediate warning indication of potential degradation, even before the user is likely to feel their symptoms [27].

SPYDR is a self-contained, earcup-mounted device with a reflectance PPG sensor embedded within the ear-seal (Figure 1). SPYDR captures, processes, and records data with the intent of identifying immediate human performance degradation without requiring any external modifications to any operational equipment. In addition to real-time recording and processing of biological data streams, the device also simultaneously measures environmental conditions with built-in sensors, currently of pressure, temperature, and acceleration, but it is designed to be adaptable to adding additional sensors as well, such as for inhalation and exhalation. When SPYDR is unplugged from its charging locker or computer system, it automatically goes into operational mode and begins recording data. When the job is finished and the device plugged back in, it automatically returns to dormant mode, downloads all data, charges the battery, and sets the clock. There is a hard power switch intended for long-term storage, shipping, and other situations necessitating the device to be powered off.

Signal processing within SPYDR was enhanced and optimized with novel, proprietary built-in algorithms specifically designed to detect and counter motion-artifact. In addition, several features of SPYDR’s design and anatomical measurement site are important contributing factors to its accuracy [26]. SPYDR’s custom-fabricated ear-cup design is an innovation to PPG sensor platforms, in that it provides adequate pressure to hold the sensor in tight yet comfortable contact with the skin while additionally blocking out ambient light that could throw off readings. This form factor alleviates a number of issues with sensor application and contact with the skin that commonly led to reduction in accurate data collection in other PPG devices, while also allowing the entire system to be a self-contained unit, housing the PPG sensor, additional environmental sensors, signal processing unit, battery, and solid-state data storage.

The anatomical location of the SPYDR sensor, firmly anchored on the head over the mastoid process, behind the ear between the pinna and the hairline, over the posterior auricular artery, is another key factor to the speed and accuracy of its readings [26]. This site has no muscle between the skin and rigid underlying bone structure, thus decreasing reading interference and/or inaccuracies created by movement or flexing of muscular tissue or vasoconstriction, common in other PPG sensor locations [18,28,29]. Additionally, at this site, SPYDR measures blood oxygen saturation of a major vessel which supplies blood directly to the brain [30]. Thus, the placement of the SPYDR sensor can most closely measure oxygenation levels of blood going to the brain, allowing it to collect data on important physiological changes in blood oxygenation levels faster than at periphery mounted devices, such as other most common PPG sites of the wrist, fingers, earlobes, ear canal, or forehead [21]. In a previous study, SPYDR identified drops in SpO2 faster than finger and forehead sensors, largely due to this behind the ear location [26].

This study seeks to validate the accuracy of the SPYDR PPG sensor system against ECG and clinical grade PPG sensor controls at the finger and forehead for accuracy in monitoring in the simulated low-motion but high-risk, high-altitude, high-pressure environment of a hypobaric chamber to simulated altitudes of 25,000 feet, and in the high-motion, high-acceleration environment of a human-rated centrifuge up to 9 G acceleration with United States Navy test pilot subjects. SPYDR is shown here to provide readings comparable to controls in the low-motion environments yet far outperforms controls in high-motion, high-acceleration environments, where PPG controls were unable to collect data and ECG results were full of extreme motion-artifacts, not being capable of collecting detailed short-term changes in heart rate in the centrifuge, which were seen with SPYDR. Results demonstrate SPYDR should be considered for a reliable system in continuous physiological monitoring for high-risk, high-motion environments, yielding a novel method for collecting data important for investigating currently inaccessible parameters of human physiology.

## 2. Materials and Methods

### 2.1. Safety Approval

SPYDR (Spotlight Labs, Haddonfield, NJ, USA) has been previously found to be FDA compliant, exceeding standards for reflectance oximetry for test subjects of all genders, ethnicities, and skin tones [26]. All testing was conducted under a protocol approved by the Ethics Committee of the United States Air Force Research Laboratory. All subjects gave their informed consent for inclusion before they participated in the study. The study was conducted in accordance with the Declaration of Helsinki, and the protocol was approved by the Ethics Committee of the United States Air Force Research Laboratory Institutional Review Board (IRB) Protocol FWR20170114H v3.00 “Multi-Channel Developmental Sensor Evaluation”—approval date, 7 May 2019.

### 2.2. Subjects and Devices

Subjects were drawn from a pool of rated United States Navy aircrew human research subjects previously identified, screened, and enrolled in the study by an established Institutional Review Board. Each subject was fit with a new, HGU-68/P flight helmet (Gentex Inc., Carbondale, PA, USA) with SPYDR devices installed and additional control sensors. For all tests, installation of SPYDR was demonstrated first by an engineer from Spotlight Labs. After the demonstration, future installations of SPYDR were conducted by US Navy personnel and contract support staff. Installation lasted fewer than 5 min per helmet and fitting of the aircrew lasted fewer than 5 min, on average. For comparative analyses controls, test subjects were also instrumented with a Masimo forehead sensor (Masimo Corp, Irvine, CA, USA), a Nonin fingertip pulse oximeter (Nonin Medical, Plymouth, MN, USA), or a clinical 12-lead electrocardiogram (ECG). For the ECG, subject’s skin was prepared by wiping the chest area thoroughly with skin cleansing (alcohol) swabs to remove any oil that may be on the skin and which can cause drift in your ECG. For the ECG, there were two groups of electrodes: six chest electrodes (see right) and four limb electrodes. The four limb electrodes, placed on the wrists and ankles, provide the electrical information that produces the six limb leads on the ECG. All devices used in the hypobaric and centrifuge tests were outfitted by a third party member of the United States Government, following initial instructions by Spotlight Labs.

### 2.3. Hypobaric Chamber Testing

A total of five test subjects (3 males, 2 females) were used for five trials. Test subjects were fitted with an HGU-55/P flyer’s helmet (Gentex Corp, Carbondale, PA, USA) retrofitted with SPYDR earcups. Subjects were also instrumented with the commercially available, FDA-approved finger pulse oximeter as well as the commercially available, FDA-approved forehead PPG sensor. Subjects were seated in an F-18 ejection seat mockup and restrained by a five-point harness inside a hypobaric chamber. Subjects were then blindly subjected to one of two hypobaric chamber profiles. Profile #1 began with a 15-min period at “sea level” before “climbing” at 5000 feet per minute to 10,000 feet equivalent pressure altitude. This pressure altitude was maintained for 10 min before climbing to 14,000-feet equivalent pressure altitude. These conditions were maintained for 10 min before climbing to a final altitude of 17,500 feet equivalent pressure altitude. Test subjects remained at 17,500 until meeting termination criteria. During profile #1, the test subjects were breathing 21% oxygen (normal atmospheric) for the entire test. Hypobaric chamber profile #2 similarly began with a 15 min period at “sea level” before climbing to 10,000 feet. Subjects then had a 10 min hold before being administered 100% oxygen for 30 min. After the 30 min oxygen prebreathe, simulated altitude was increased to 25,000 feet equivalent pressure altitude. The subject was then given 21% oxygen (ambient) mixture and held at this configuration until meeting termination criteria. Termination criteria: If at any point one of these criteria were met, the test was terminated immediately, subjects were administered 100% oxygen, and the chamber was returned to ground level at 5000 feet per minute: (1) test subject called terminate for any reason, (2) any member of the test administration team called terminate, (3) SpO2 by finger probe was less than 60% for 10 s, or 4) time-limit at altitude exceeded 20 min.

### 2.4. Centrifuge Testing

Total of five test subjects (3 male, 2 female) were identified; however, only two trials were conducted (1 male, 1 female). SPYDR earcups were installed and fitted into an HGU-55/P flyer’s helmet (Gentex Corp, Carbondale, PA, USA), as described above. Centrifuge test subjects were also instrumented with both finger and forehead control PPG sensors as well as a 12-lead electrocardiogram (ECG). Finally, subjects had their extremities wrapped with compression bandages and were outfitted with a Full-Coverage Anti-G Suit (FCAGS), restraining harness, and a flyer’s oxygen mask. After being seated in the centrifuge, subjects were restrained to the seat by a standard centrifuge restraining harness. With their helmet on and mask down, connected by a single bayonet on the left side of the helmet, but not secured in place, subjects had a five-minute rest period with the centrifuge stopped. The oxygen mask was connected to the centrifuge regulator, and communications were established with the test administrator and safety observer. Subjects were then instructed to raise and secure their masks, and the centrifuge was brought to idle speed (1.6–1.8 Gz, nominally). Test subjects were then subjected to four high-G profiles designed to simulate Anti-G Straining Maneuvers (AGSM) training as prescribed by NATOP and USAF instructions. In between each profile, the centrifuge was brought to a full-stop, and subjects could remove their oxygen masks. Profile 1. Subjects were exposed to a gradual onset acceleration (approximately 6 G/minute onset rate) until reaching 9.0 Gz. Profile 2. Subjects reinstalled their oxygen masks before the centrifuge was returned to idle speed. Subjects then experienced 5 Gz for 30 s. After 30 s at 5 Gz, the centrifuge was returned to “idle” speed, and subjects lowered their oxygen masks for a 2 min break. After 2 min, subjects experienced 7.5 Gz for 20 s. The centrifuge was again slowed to idle speed, and subjects lowered their masks for another 2 min break. Subjects replaced their masks and were then exposed to 9 Gz for 15 s. The centrifuge was stopped, and subjects lowered their masks. Profile 3. Air Combat Maneuvering profile, after raising their masks, subjects experienced multiple, repeated, rapid-onsets to high-Gz for approximately 20 s before unloading to 2 Gz for 10 s. The specific acceleration profile was: 7.5, 2, 7, 2, 6.5, 2, and 6 Gz. After the final 6 Gz exposure, the centrifuge was stopped, and subjects lowered their masks before completing another 5 min cognitive test. Profile 4. Subjects reinstalled their masks before the centrifuge was accelerated to idle speed. Subjects then experienced 5 Gz for 30 s. After 30 s at 5 Gz, the centrifuge was returned to “idle” speed, and subjects lowered their masks for a 2 min break. After 2 min at 18 Gz, subjects experienced 7.5 Gz for 20 s. The centrifuge was stopped, and subjects lowered their oxygen masks.

### 2.5. Data Reporting and Analysis

Raw data from SPYDR were compiled as a comma-separated variable worksheet (.CSV) and emailed securely transmitted to the test team government representative and their designated US government-appointed representatives. Data obtained from the government’s laboratory instruments were similarly compiled and made available to Spotlight Labs. Important to note, because the control devices recorded data at 0.2 Hz (once every five seconds), whereas SPYDR records at 1 Hz (once per second), the devices each captured data at different intervals, and all data were first reduced to 5 s (0.2 Hz) sample rates, dropping additional SPYDR data. To account for beat-to-beat variability and device-specific signal processing differences, a six-measurement rolling average window was calculated used to compare SPYDR with control devices (30 s window).

## 3. Results

### 3.1. Hypobaric Chamber Results

Hypobaric chamber tests were run to determine that SPYDR can provide accurate and reliable data for blood oxygen saturation (SpO2) and Pulse Rate (PR) in reduced atmospheric pressure. Tests were conducted comparing SPYDR to a commercially available, FDA-approved, finger-mounted pulse oximeter. Pulse rate as determined by SPYDR evaluated against the control device by Pearson’s correlation coefficient and Bland–Altman analysis. For all tests, SPYDR’s barometric pressure sensor matched the hypobaric chamber’s control sensor with over 99.99% accuracy (not shown). While the hypobaric chamber tests presented herein only go to 25,000 feet, SPYDR has been previously tested to 100,000 feet. Capture rates of 100% from SPYDR were observed for all users and all tests except for hypobaric chamber test #4. Approximately 20 min into the test, the test subject (not rated aircrew) complained of a hotspot due to the SYNWIN earbuds. As the subject adjusted the helmet to alleviate pain, the SPYDR sensor became disabled, resulting in reduced capture rates. Until that point, data capture in all tests was 100%, in large part due to the redundant two ear-cup system for filling in any missing datapoints.

US Government control data for blood oxygen saturation (SpO2) and Pulse Rate (PR) were provided only for the first two hypobaric chamber tests. The control sensor used was a commercially available, FDA-approved, finger-mounted pulse oximeter. As shown in Figure 2, SPYDR’s SpO2 reading matched the control sensor with a correlation coefficient of 0.92 and 0.89, respectively. Correlation coefficients are reduced because SPYDR responds to changing physiological parameters much faster than peripheral devices. This is clearly seen in the graph on the right in Figure 2, where the test subject experienced a severe case of hypoxia. SPYDR detected this condition and would have issued an audible alert up to one full minute before the subject lost consciousness. We do note a discrepancy in the tightness of tracking between SPYDR and the PPG finger sensor, and we believe the differences noted are based on several factors. One main issue is that the commercial device is reporting data once every 5 s, measured at the fingertip, and is well known to be susceptible to motion artifact. The SPYDR device is reporting data once every second and is less susceptible to motion. We had to average the SPYDR data in time (every 5 s) to match up to the fingertip probe. Also worth noting is that the control PPG sensor is reading on an extremity, on the finger, and the SPYDR device is read from the head, behind the ear, in direct line from the heart to the brain, and there are many differences in perfusion between these two locations and between the physiology of different people. In addition, SPYDR is a reflectance PPG sensor, and the finger sensor is transmission mode. However, this was not a study for the accuracy of SPYDR compared to the finger PPG sensor, as that has already been validated in hypoxic environments and using the gold-standard Arterial Blood Gas test in our previous study [26].

In Figure 3, Pulse Rate as determined by SPYDR was plotted against the control device and correlation coefficients were determined by Pearson’s method. Pearson’s method is the most accurate when the range of data tested have a high degree of variability. The first was relatively benign, and as such, the test subject’s heart rate did not vary widely (77 +/− 7 BPM). The second test was much more physiologically demanding, resulting in a much broader range of heart rates (90 +/− 30 BPM). Pearson’s correlation for the second test was 0.93 (Figure 3). For both tests at profile #2, after a short time (6.6 min, 4.5 min) at 25,000 feet, the subjects became severely hypoxic and required a rapid descent with 100% oxygen recovery.

We observed a slight decrease in correlation coefficients for SpO2 in the hypobaric chamber in both tests because, as shown in a previous study, SPYDR responds to changing physiological parameters faster than peripheral devices, based on SPYDR sensor reading location in direct line from the heart to the brain, as opposed to a slower SpO2 change recording at the peripheral sensor on the finger. This is clearly seen in the right panel of Figure 2, where the test subject experienced a severe case of hypoxia and sudden loss of consciousness. SPYDR detected this condition and would have issued an audible alert up to one full minute before the subject lost consciousness under normal operating programing, disabled for this test.

Figure 4 shows Bland–Altman plots of PR and SpO2 measurements comparing SPYDR with PPG readings at the finger and forehead. The x-axis and y-axis indicate the mean and difference of the PPG-derived and the corresponding reference respiratory frequencies, respectively. The *y*-axis represents the difference between the two signals, and the *x*-axis represents the average of the two measures. The center dotted lines show the mean bias of 1. The limits of agreement (LoA) are mean ±4.84 SD. The main source of error was that the control device only records once every 5 s, whereas SPYDR records every second; therefore, syncing clocks was a challenge. For the Bland–Altman plot, there are 10,000 data points plotted. The two standard deviation lines show that 96% of data is +/− 10 bpm of the mean, which is within the United States FDA requirements for PPG sensor accuracy.

### 3.2. Human Centrifuge Testing Results

Subjects were tested in a human centrifuge to evaluate the stability and continuity of SPYDR in a dynamic, varied, high-acceleration environment, analogous to combat flight. Subjects were instrumented with a SPYDR-equipped helmet and commercially available, FDA-approved electrocardiography sensor. It is important to note that PPG finger and forehead control sensor data were unavailable following the centrifuge tests due to high-motion artifact. Originally designed to be a paired study with five hypobaric chamber tests and five centrifuge tests, the remaining three tests were cancelled because SPYDR outperformed expectations, and there was lack of control data for evaluation. SPYDR had a 100% data capture rate and shows a much more reliable and usable pulse rate signal, suggesting that PR determined by PPG is advantageous than ECG in high-acceleration environments. Data capture rates for PR and SpO2 as measured by SPYDR was 100% for both centrifuge trials, including multiple periods of 9 G acceleration (Figure 5). It is important to note, that the aim of this work was not about comparing device A vs. device B for each individual subject; it is about number of exposures to high-motion environments. Therefore, each person did 12 exposures, × 2 people, and then we have n = 24 exposures of high motion, which we believe is sufficient for this type of study. Both tests put the human subjects in highly physiologically challenging and risky environments, at the extremes for both low oxygen and highest forces (9 Gz). For the centrifuge tests in particular, subjects must be trained properly, as ours were for this study, following our IRB protocol, in how to breathe and contract their muscles to keep the blood from pooling in their extremities so as to not pass out from blood draining out the head, although even healthy and highly trained United States fighter pilots still regularly pass out in these centrifuge test profiles. Therefore, we felt it was unnecessary to continue to expose human subjects to these conditions beyond what we needed for our data collection.

Acceleration as measured by SPYDR’s in-unit accelerometer sensor was compared to the acceleration computed by the centrifuge (Figure 6). The decreases in blood oxygen saturation (SpO2) and increases in pulse rate commensurate with high-acceleration exposures were notable, as would be expected, highlighted by the red arrows in Figure 6. The presence of a pulsatile waveform suggests that blood is flowing, yet decreased oxygen saturation indicates reduced perfusion which typically pre-empts a G-induced Loss of Consciousness (G-LOC).

Centrifuge tests in this study simulated extremes of high-acceleration forces for humans, although forces commonly experienced during tactical flight maneuverings. As part of SPYDR’s environmental monitoring system, its built-in accelerometers tracked acceleration in relation to SpO2 and PR changes in the subjects (Figure 7). As also shown in Figure 7, SPYDR’s onboard accelerator matched the accelerometer in the seat of the centrifuge. In demonstration of SPYDR’s onboard accelerometer’s accuracy, SPYDR’s measured acceleration is roughly 9% lower at the head as compared to the centrifuge’s seat, as would be expected as the subject’s head is 9% closer to the center of rotation of the centrifuge than the seat.

## 4. Discussion

SPYDR represents the first self-contained PPG sensor and warning system for routine physiological and environmental monitoring in extreme high-motion and high-altitude environments. To date, there has been a large unmet need for accurate, reliable methods for routine monitoring of physiological metrics of humans in high-motion, high-risk environments, especially those of tactical flight for which SPYDR was optimized, where many accidents occur due to the challenges of extreme environmental conditions on the human body. The SPYDR PPG sensor device is an important advancement to current biomedical monitoring devices, in that it is shown here to capture data in an environment of extremes of high motion and high force seen in tactical aviation which poses high risks to human health, a human-rated hypobaric chamber, and centrifuge. While the SPYDR device is further validated for the first time to work as well as PPG sensor controls in detecting changes in pulse rate and blood oxygen saturation in the hypobaric chamber in this study, it is an important finding that, as high motion is introduced in the centrifuge studies, PPG sensors at the finger and forehead were unable to capture any usable data. Although ECG was able to capture some data, it is visually clear from the figure graphs that ECG’s data were, like the PPG sensor controls, also largely unusable due to extremely high levels of motion artifact, whereas it is clear that SPYDR was able to continuously monitor physiological parameters, despite the extreme levels of motion. The demonstration of SPYDR’s ability to tightly track accurate physiological data in such an extreme of a high-motion environment represents a huge advancement for the field of PPG sensor technology that we believe warrants the publication of this study.

As demonstrated by the extremely high-motion artifacts in the centrifuge (Figure 5) and the inability of the PPG sensor controls to capture any usable data, there is currently no other device or technology capable of capturing such a broad range of physiological metrics correlated directly with environmental parameters. Additionally, SPYDR provides a unique, novel real-time warning system for physiological decrements in high-risk, high-motion environments [20,21]. Increasing our currently limited understanding of human physiology in dynamic environments with the introduction of this technology into every-day use is a logical first step for quantifying, understanding, predicting, and preventing human error mishaps in high-risk operational environments.

It has become more commonplace to see ECG routinely used during periods of motion, but it is typically only accurate enough to identify QRS complex to determine heart rate but not the full P-QRST-T ECG wave complex with each heartbeat. As ECG measures microvoltage potentials, which increase with every muscle movement, ECG measurements that claim to be able to measure through motion cannot be completely accurate. They typically just ignore motion artifacts where they occur. During periods of high-motion artifact in ECG measurements, the averaging window is generally large, and the data still remain very noisy. Noisy data are typically then dropped or not reported, as they are averaged out over a large window of time, utilizing data when there are periods of no motion for overall readings. For example, according to the Bruce protocol, heart rate and rating of perceived exertion are taken every minute, and blood pressure is taken at the end of each stage (every three minutes). This study compares the ability of ECG to capture data through periods of known, simulated, extremes of high motion, compared to SPYDR’s ability to record data every second. This is important as SPYDR’s physiological monitoring capabilities every second are connected to an aural warning system for physiological decrements that fall below a set threshold. In the extremes of speed and motion of tactical flight in military jet aircraft that SPYDR was optimized for, a jet can go from above 30,000 ft to crashing into the ground in 20 s if the pilot is even momentarily impaired in proper cognitive functioning and changes the level course of the jet, which has happened. These occurrences are believed to be due to cases of hypoxia, although the exact physiological mechanisms are unknown because of lack of capable physiological monitoring in those environments, a need in the field we are seeking to address with SPYDR. While useful for use in routine heart rate monitoring situations, such as a stress test, ECG is incapable of capturing data every second during periods of high-motion, as SPYDR is shown capable of, and therefore, ECG should not be considered a reliable device for monitoring physiological data in critical cases of extremes of motion, such as piloting a jet aircraft, for which SPYDR was specially designed.

Signal disruption in the controls were consistently observed in this study with periods of dynamic movement and motion (Figure 5). ECG especially was completely unusable during these periods, strongly demonstrating the unreliability of ECG during periods of high motion. This is due to a number of electrophysiological factors and highlights the general unacceptability of using ECG in environments of movement, especially tactical aircraft. ECG has been found to be highly unreliable and mostly unusable during periods of motion [16]. Also problematic, ECG technology captures data through electrodes, which are too cumbersome and distracting to wear while working, especially in a high-stakes environment such as flight, defense, or first response.

Electrocardiography artifacts are defined as electrocardiographic alterations not related to cardiac electrical activity and are known to be frequent. Motion artifacts in ECG can include external and internal interference caused by numerous factors such as shaking with rhythmic movement, poor grounding of the device, interference by other devices in the vicinity such as electrical beds, surgical and fluorescent lamps, artifacts produced by alternating current affecting the ECG baseline, mistaken placement in the cable junction box, muscle twitching, inappropriate cleansing of the skin, excess of precordial conductive gel, and mistaken placement of both limb and precordial leads, all can all cause irregularities in an ECG baseline. as [16]. As a result of any or a combination of these potential artifacts, the components of the electrocardiogram (ECG) such as the baseline and waves are distorted, as seen in this study. These factors all highlight general unacceptability of using ECG to regularly and reliably collect physiological data in dynamic environments, especially those of tactical flight [19]. By contrast, SPYDR was shown to accurately detect physiological decrements, such as low SpO2, in all periods of high motion in all subjects.

Results of this study demonstrate the capability of providing accurate biodata from a PPG sensor for all users regardless of gender, skin pigmentation, or hair length. Throughout the duration of the test, there were no reports of abnormal diaphoresis or movement of the device within the helmet. No user reported any discomfort from SPYDR, although other sensors and peripheral test equipment did present some discomfort/hotspots for two test subjects. In the centrifuge tests, SPYDR indicated decreased blood oxygen saturation of the subject at periods of high acceleration. In high-acceleration environments of tactical flight, peripheral sensors cannot be used effectively because of blood pooling and altered blood pressure at the extremities. [31].

A limitation of the statistical analysis of the collected data was that the control PPG sensors only recorded data once every five seconds, while SPYDR captures data at a rate of once every second; therefore, SPYDR data had to be adjusted accordingly. Very rapid transient perturbations in physiological data, such as spikes in heart rate or drops in blood oxygen saturation, are missed when recording only once every five seconds by other devices. Thus, synchronizing and correlating SPYDR’s data to the control data set were often difficult and yielded a showing of less accurate results for SPYDR with reduced statistical significance.

The device was designed with capabilities to be easily augmented and individualized for a variety of scenarios, making it highly adaptable for various industries [9]. Expanded capabilities and utilization of SPYDR could be beneficial in other fields, such as sports, mental health care, and medicine, to detect and alert any user to the potential for an abrupt physiological or cognitive change [9]. The development of alerting algorithms based on these data streams in the future could potentially provide users with key information and pre-emptive warning about impending emergencies, which have historically led to loss of life and destruction of property, as the system could provide a critical warning capability, alerting subjects or supervisions to developing situations and impending emergencies long before they detect any perceived degradation.

Additional future implications are profound for industries where employees are engaging in activities that could become risky in the presence of human errors, which the personnel are currently on their own to detect, such as emergency response, sports, medicine, mining, and transportation. Personnel in these extreme physical and mental operational environments have a need for an objective way to measure their ability to perform missions effectively and safely for both themselves and others when they reach the point of being unable to assess it for themselves. These errors can cause injury or death to the individual and passengers, and the environmental impact associated with events such as plane crashes, oil spills, and train derailments can be catastrophic.

In conclusion, both simulated and in-flight test results of this study confirm that SPYDR’s real-time physiological and environmental monitoring accuracy rates exceed previously fielded technologies in extreme operational high-motion environments. Future data gathered by SPYDR through extensive in-flight testing would be extremely valuable in understanding and developing solutions for underlying causes of physiological incidents in many varied venues of extreme mental and physical operational environments. The study confirms that SPYDR should be considered for routine use in extreme operational environments, especially tactical flight, and expanded research into the potential for data mining and enhanced biometric quantification of human performance and cognition should be explored.

## Figures and Tables

**Figure 1 sensors-21-04543-f001:**
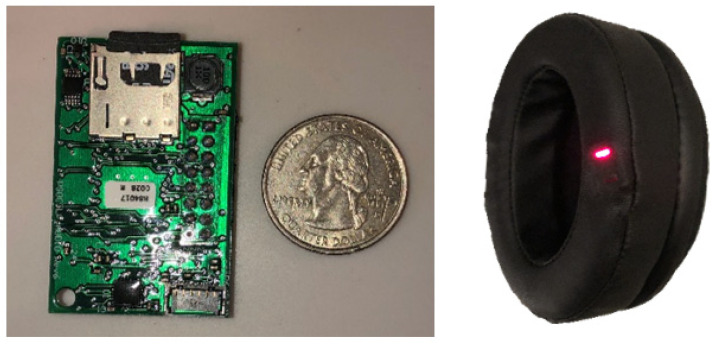
The SPYDR earcup system is a self-contained unit that can act as a functional replacement for earcup in standard flight helmets or headsets. Photograph of SPYDR’s internal processor and circuit board (**left**) and assembled SPYDR earcup (**right**) with PPG sensor embedded within the earseal at the red light.

**Figure 2 sensors-21-04543-f002:**
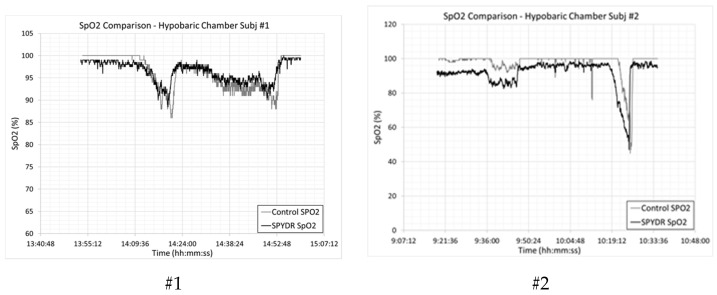
Comparison of blood oxygen saturation (SpO2) as measured by SPYDR vs. control finger PPG sensor. Note that SPYDR responds to changing physiological conditions faster than the peripheral finger sensor, especially noticeable during a period of acute hypoxia of Subject #2 in the right panel, supporting findings from a previous study. Correlation by Pearson was 0.92 for test #1 and 0.89 for test #2.

**Figure 3 sensors-21-04543-f003:**
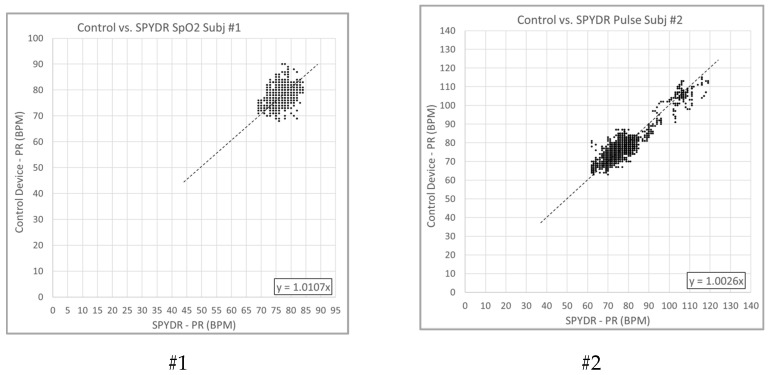
Correlation plot of Pulse Rate (PR) as determined by finger PPG sensor control vs. SPYDR for test #1 (**Left**) and test #2 (**Right**). Correlation coefficient by Pearson was 0.78 for test #1, 0.93 for test #2.

**Figure 4 sensors-21-04543-f004:**
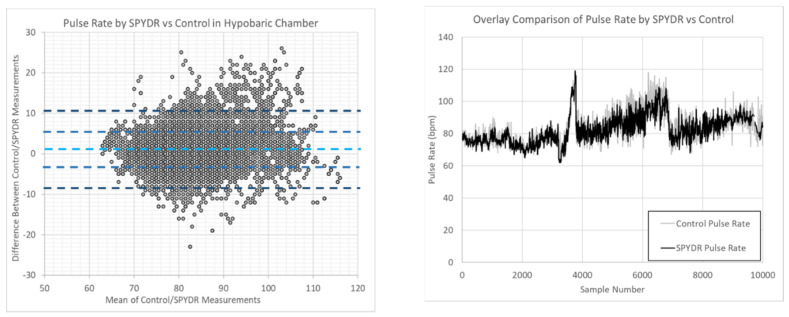
Figure on the left is a Bland-Altman plots of PR measurements comparing the agreement of the systems comparing SPYDR with control finger and forehead PPG sensors for all subjects. The panel on the right is the same data over time, adjusted for the control device’s reduced capture rate.

**Figure 5 sensors-21-04543-f005:**
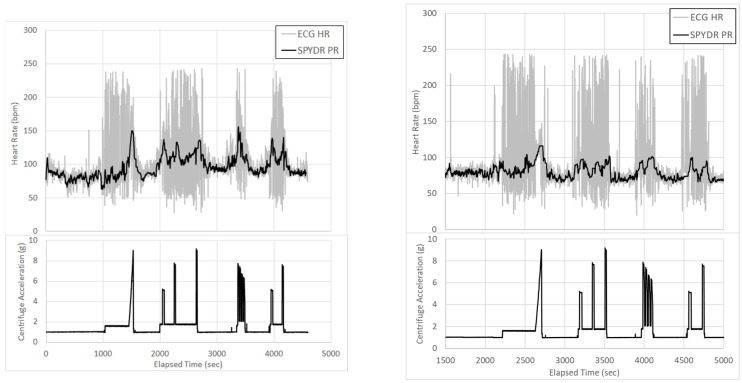
Centrifuge testing of heart rate as determined by electrocardiogram of a test subject in a manned centrifuge simulating high-acceleration flight. ECG was highly unreliable and mostly unusable, especially during periods of motion and acceleration.

**Figure 6 sensors-21-04543-f006:**
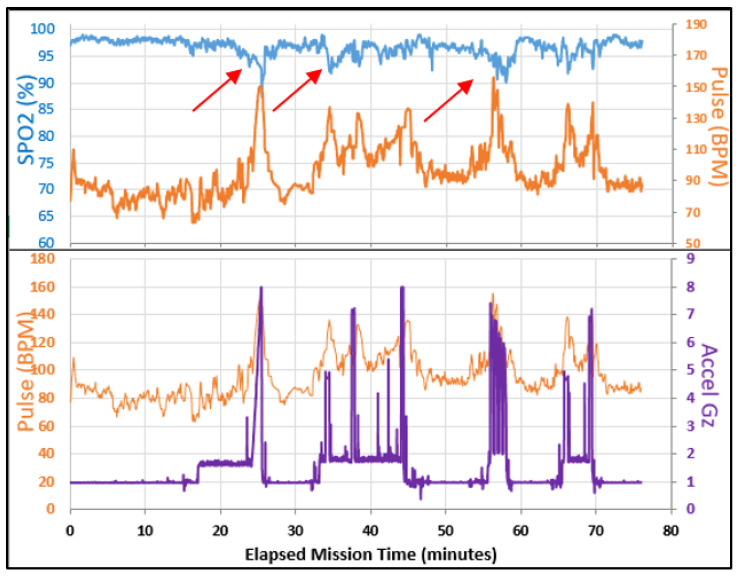
Manned-centrifuge full data as reported by SPYDR for a test subject simulating high-acceleration flight. Note decreases in blood oxygen saturation (SpO2) and increases in pulse rate commensurate with high-acceleration exposures (red arrows). The presence of a pulsatile waveform suggests that blood is flowing, yet decreased oxygen saturation indicates reduced perfusion which typically pre-empts a G-induced Loss of Consciousness (G-LOC).

**Figure 7 sensors-21-04543-f007:**
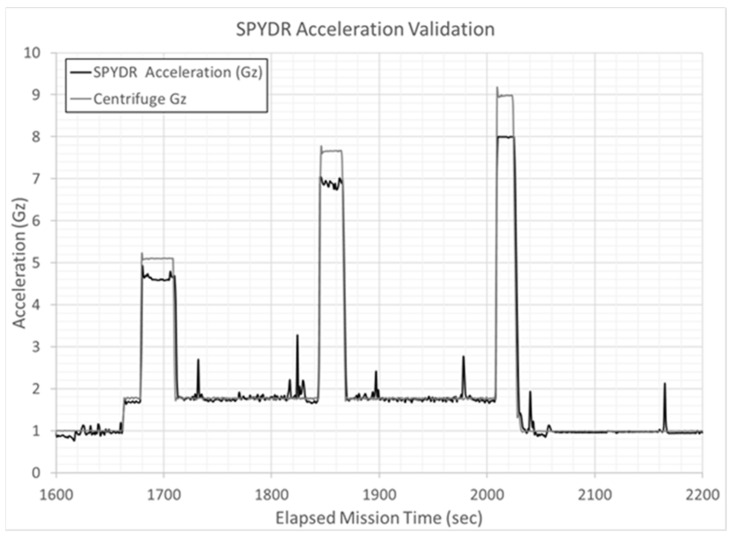
Acceleration as measured by SPYDR was compared to the acceleration computed by the centrifuge. Interestingly, the centrifuge accelerometers are located at the seat, whereas SPYDR is measuring acceleration at the head. Since the user’s head is 9% closer to the center of rotation of the centrifuge than the seat, acceleration is roughly 9% lower at the head as compared to the acceleration sensed by the centrifuge.

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
