# Peer review of "Photoplethysmography behind the Ear Outperforms Electrocardiogram for Cardiovascular Monitoring in Dynamic Environments"

_sensors, 2021, doi:10.3390/s21134543_

Round 1
Reviewer 1 Report
- In line number 64 ~ 71, ECG is very sensitive to motion artifact. But, I don't think it's true anymore. Many studies to overcome the motion artifact have been reported as below. Therefore, ECG is measured in the Bruce protocol (https://en.wikipedia.org/wiki/Bruce_protocol).
https://ieeexplore.ieee.org/abstract/document/7933339
https://www.mdpi.com/1424-8220/14/8/14732
- The detail of ECG equipment that you used in the experiment is not mentioned.
- On the right side in the figure2, as you mentioned we can see the faster changes clearly but SP02 values does not accurate all the time. On the other hand, on the left side in the figure, the faster changes is not observed and SpO2 values are more accurate. What makes this different?
- On the left side in the figure 3, I doubt the value of correlation coefficient. I can’t see any correlation between pulse rate from SPYDR and pulse rate from control device. The dots in the scatter plot form an almost circular shape. The gradient of the regression line can’t be almost 1 in this case.
Reviewer 2 Report
Authors compare the effectiveness of their SPYDR sensor with standard vital signs measurements devices. The topic is very interesting and the proposed approach seems to be very innovative. However two are the main concerns about this work:
1) SPYDR device has been already described in a previous work [26]. Authors must highlight what is the difference between this article and the previous one.
2) The aim of this paper is to compare the effectiveness of their device with standard devices. For this reason measurements in extreme conditions have been carried out. However, for the first experiment only five subject have been considered, and for the second one, two. In my opinion the number of considered subject is not sufficient to derive general observation on the effectiveness of the proposed device. Moreover, figure 4 shows that the two measurements do not agree, and several points are out of the level of acceptance. Finally, measurements for different subjects must be considered in the same graph to observe how well the device is able to measure vital parameters.
The quality of the paper is high, but more experiments are needed.
Reviewer 3 Report
This is a clearly written paper that presents applications of SPYDR (Standalone Performance Yielding Deliberate Risk) earcup device for monitoring of cardiovascular data in high motion and high-risk environments.
Major drawbacks of the paper are: 1) small number of subjects (1 male and 1 female) that participated in the experiment and 2) unsystematic approach in presenting accuracy results for different experiment profiles (also, it is not clear what is the overall time of the experiment protocol). Suggestions for paper improvements are: 1) presenting the experiment protocol visually (e.g. using a timeline), 2) increasing number of participants, 3) systematic presentation of error analysis and SPYDR advantages (e.g. using a table).
Round 2
Reviewer 2 Report
The article has been improved according with the reviewers' comments.
Reviewer 3 Report
The authors highlighted sufficiently the main contributions of their work and they have improved the paper.